# Field-warmed soil carbon changes imply high 21<sup>st</sup> century modeled uncertainty

Katherine Todd-Brown[1], Bin Zheng[1], Thomas Crowther[2]

[1]Pacific Northwest National Laboratory, Richland, WA, 99354, USA
5  [2]Institute of Integrative Biology, ETH Zürich, Univeritätstrasse 16, 8006, Zürich, Switzerland

*Correspondence to*: Katherine Todd-Brown (katherine.todd-brown@pnnl.gov)

**Abstract.** The feedback between planetary warming and soil carbon loss has been the focus of considerable scientific attention in recent decades, due to its potential to accelerate anthropogenic climate change. The soil carbon temperature sensitivity is traditionally estimated from short-term respiration measurements -- either from laboratory incubations that are artificially 10  manipulated, or field measurements that cannot distinguish between plant and microbial respiration. To address these limitations of previous approaches, we developed a new method to estimate soil temperature sensitivity ($Q_{10}$) of soil carbon directly from warming-induced changes in soil carbon stocks measured in 36 field experiments across the world. Variations in warming magnitude and control organic carbon percentage explained much of field-warmed organic carbon percentage ($R^2$=0.96), revealing $Q_{10}$ across sites of 2.2, [1.6, 2.7] 95% Confidence Interval (CI). When these field-derived $Q_{10}$ values were 15  extrapolated over the 21<sup>st</sup> century using a post-hoc correction of 20 CMIP5 Earth system model outputs, the multi-model mean soil carbon stock changes shifted from the previous value of 88 ±153 Pg-carbon (weighted mean ± 1 SD), to 19±155 Pg-carbon with a $Q_{10}$ driven 95% CI of 248±191 to -95±209 Pg-carbon. On average, incorporating the field-derived $Q_{10}$ values into Earth system model simulations led to reductions in the projected amount of carbon sequestered in the soil over the 21<sup>st</sup> century. However, the considerable parameter uncertainty led to extremely high variability in soil carbon stock projections within each 20  model; intra-model uncertainty driven by the field-derived $Q_{10}$ was as great as that between model variation. This study demonstrates that data integration should capture the variation of the system, as well as mean trends.

## 1 Introduction

The flux of carbon dioxide between the soil and atmosphere is a major control on atmospheric carbon dioxide concentrations. Warming temperatures, driven by increases in atmospheric carbon dioxide, have the potential to stimulate carbon 25  decomposition, accelerating its release into the atmosphere (Davidson and Janssens, 2006). If this is not counterbalanced by an equivalent increase in primary productivity (the opposing carbon flux) then it has the potential to drive a land carbon-climate feedback that will accelerate anthropogenic climate change. Recent global compilations of data from ecosystem warming experiments lend support to this idea (Carey et al., 2016), suggesting that warming alone could drive a loss of carbon from the upper soil horizons (Crowther et al., 2016). However, these studies addressed the impact of warming in isolation, and

it remains unclear how this process will interact with the variety of other global change drivers to affect the global soil carbon stock over the rest of this century. Reflective of such uncertainty, soil carbon changes projected for 2100 under business-as-usual scenario for Coupled Model Intercomparison Project Phase 5 (CMIP5) range from -70 to 250 Pg-carbon across different Earth system models (Todd-Brown et al., 2014), making the land-carbon feedback one of the largest sources of uncertainty in future climate projections (Friedlingstein et al., 2014). Improving the soil carbon component of the Earth system models is essential to predicting the future evolution of the Earth system and thus establishing meaningful greenhouse gas emissions targets.

A fundamental parameter describing soil temperature-sensitivity in soil carbon models is the $Q_{10}$ – the factor of the change in decomposition rate associated with 10°C of warming from a reference temperature (Davidson and Janssens, 2006; Lloyd and Taylor, 1994). Traditional laboratory incubations have found a wide range of $Q_{10}$ values, varying from 1.4 (Townsend et al., 1997) to > 3 (Davidson et al., 1998, 2006) with 2 being the most commonly accepted value. Complicating this, theoretical analyses based on chemical kinetics suggest $Q_{10}$ is itself dependent on temperature (Davidson and Janssens, 2006; Lloyd and Taylor, 1994), though these values are typically very close to 2 in most environmental temperature ranges (Lloyd and Taylor, 1994). More recently, large-scale analyses of field respiration converge on $Q_{10}$ estimates of 1.4 to 1.5 (Bond-Lamberty and Thomson, 2010; Hashimoto et al., 2015; Mahecha et al., 2010). Unsurprisingly, this temperature response is also critical in Earth system models, where the temperature sensitivity parameter is known to be a major driver of variation (Booth et al., 2012; Jones and Cox, 2001; Jones et al., 2006). However, it is unclear what is driving the lower $Q_{10}$ estimates in these field-based syntheses compared to the average lab-based estimates from single-site studies, and there appears to be a relatively wide range of 'typical' $Q_{10}$ values in the literature. Nevertheless, most Earth system models use values that range from 1.5 (Oleson et al., 2013; Raddatz et al., 2007) to 2 (Bonan, 1996; Cox, 2001).

Traditionally, these $Q_{10}$ values have been calculated from warming-induced changes in soil respiration rates. However, this approach has two main limitations: 1) respiration rates measured under idealized laboratory conditions fail to reflect the structure, heterogeneity and variability of natural systems, whereas 2) field measurements cannot directly isolate heterotrophic soil respiration from autotrophic root respiration without substantially altering the system. Estimating $Q_{10}$ directly from warming-induced changes in soil carbon stocks could be a valuable approach to address these limitations, but the variability and relative imprecision of soil carbon stock data necessitates a large sample size to adequately describe variation at the global scale (Bradford et al., 2016). Yet, results from a recent Earth system model meta-analysis indirectly suggests that, with enough sample coverage it may be possible to infer $Q_{10}$ directly from changes in soil carbon stocks (Todd-Brown et al., 2014).

Here we present a new approach to estimate the global $Q_{10}$ value from net changes in soil carbon stocks under warming, rather than soil respiration measurements, and examine the consequences of these estimates –with associated uncertainty– on CMIP5 Earth system model projections of global carbon storage over the rest of the 21st century. To do this, we use a global database of soil carbon stock data from 36 field-warming experiments around the world, each of which includes control (ambient) plots, and those which have been warmed for extended (years to decades) periods of time (Crowther et al., 2016) (Tables SI1) and outputs from 20 CMIP5 (Taylor et al., 2011) Earth system models RCP 8.5 business-as-usual experiment (Tables 1 and SI3).

These field data were used previously to derive Earth system model independent estimates of global soil carbon temperature sensitivity where the effect of warming was isolated from other global change drivers or the interacting climate system (Crowther et al., 2016). In this study we develop a novel approach that enables us to explore these field results in the context of the temperature sensitivity function ($Q_{10}$) used in integrated Earth system model. We then examine the consequences of the data-driven $Q_{10}$ estimates, and the associated uncertainty, for CMIP5 Earth system model projections of global carbon storage over the rest of the 21$^{st}$ century using a novel post-hoc modification of the CMIP5 simulation outputs.

## 2 Materials and Methods

### 2.1 Field sites

The field sites were drawn from a previous analysis (Crowther et al., 2016). From this initial database of 48 paired case-control studies, we selected 36 studies that were run longer than 2 years to match the metastable state assumption articulated below. 18 of these sites were temperate grasslands, savannas, and shrublands, 10 temperate broadleaf and mixed forests, 6 tundra, 1 boreal forests or taiga, and 1 site was in a Mediterranean forest, woodland and scrub. A traditional statistical analysis of the sites is provided by Crowther *et al.* (2016). For this study, we used the increase in soil temperature due to warming, length of the study, and the percent of soil organic carbon in paired warmed and control plots (Table SI1).

### 2.2 $Q_{10}$ calculations

We calculated traditional $Q_{10}$ estimates based on these warming-induced soil carbon losses, enabling us to embed this temperature sensitivity information into a soil decomposition model framework. Traditional soil decomposition models follow a first order linear decay framework where:

$$\frac{d\boldsymbol{C}(t)}{dt} = u_{in}(t)\boldsymbol{b} - (\boldsymbol{Q_{10}}(T,t)\boldsymbol{KA})\boldsymbol{C}(t),$$ (1)

where the $\boldsymbol{C}$ is a vector of soil carbon pools with unique turnover times, $t$ time, $u_{in}$ a scaler of soil carbon inputs, $\boldsymbol{b}$ an allocation vector describing how the inputs are divided between the soil carbon pools, $\boldsymbol{K}$ is a diagonal matrix representing the decomposition rates of the pools, $\boldsymbol{Q_{10}}$ is a diagonal matrix with entries of the form $q_i^{(T(t)-T_0)/10}$ representing the temperature sensitivity factor, $T$ a scalar describing the soil temperature and $T_0$ an arbitrary reference temperature, and $\boldsymbol{A}$ the transfer matrix representing movement of carbon between soil carbon pools.

The temperature sensitivity was assumed to be constant across pools. This allows us to collapse the diagonal $\boldsymbol{Q_{10}}$ matrix to be collapsed into a single scalar value of the form $Q_{10}^{(T(t)-T_0)/10}$. This constant temperature sensitivity assumption is discussed below and follows the structure of the CMIP5 Earth system models.

In general, there are three classes of pool structure for traditional models: *independent* where there was no exchange between soil carbon pools making $\boldsymbol{A}$ the identity matrix, *cascade* where pools with faster turnover times passed carbon to pools with slower turnover times making $\boldsymbol{A}$ a lower triangular matrix, and fully *feedback* models where carbon was exchanged between

faster and slower pools and vice-versa making $A$ a fully dense matrix. In all cases $KA$ is an M-matrix, implying there exists an inverse with all positive entries. For the independent and cascade pools $KA$ is diagonalizable, implying it can be broken down into a diagonal matrix $D$ and an invertible matrix $P$ such that $KA=P^{-1}DP$.

For most well-developed soils, soil carbon stocks are at a metastable state where soil inputs approximately equal outputs (see

Results for discussion of Earth system model outputs). Given that $KA$ is an M-matrix and this metastable state approximation, we can describe the total soil organic carbon as follows:

$$C = \frac{u_{in}}{kQ_{10}^{(T-T_0)/10}},\tag{2}$$

where $C$ is the total organic carbon stock, $u$ the sum of the soil inputs, and $k$ a bulk decomposition rate that can be constructed from the decay matrix $KA$ and allocation vector of the soil inputs $b$. For details see of this derivation the SI: Mathematical

Analysis.

We can now describe the soil carbon stock difference between two soils with the same decay rate but different temperatures and inputs. This could either be two time points from a simulation where the soil output is close (within 10%) of the soil inputs, or a warmed treatment and a control:

$$C_2 = C_1 \left( \frac{u_2}{u_1} Q_{10}^{(T_1-T_2)/10} \right).\tag{3}$$

For the field sites, we assume that the relative change in inputs due to warming is negligible compared to the effect on the decomposition rate across sites and that the main driver of differences in decomposition rates between control and treatment is the warming treatment. Leading us to:

$$C_w = C_c Q_{10}^{-\Delta T/10}.\tag{4}$$

Finally, we assume that the bulk density of the soil at a given site was unaffected by the warming treatment. This allows us to

use the mass percent soil organic carbon instead of the soil organic carbon density for Eq. 4.

## 2.3 Model-data integration: parameter fitting

Given the relatively small parameter space, we choose a brute-force model-data integration approach where we iteratively calculated the predicted change in soil carbon stock given the control soil carbon (Eq. 4) across a range of $Q_{10}$ values from 0.1 to 5 in 0.1 increments. We set the lower bound of the $Q_{10}$ range to 0.1 instead of 1 for two reasons. First, while it is generally

accepted that warmer soil temperatures will increase soil respiration (constraining $Q_{10} > 1$), it is possible that a warmer soil would result in drier soils and suppress soil respiration. In addition, numerically we wanted to bracket the expected parameter range with our prior. Data-model fits were scored using root mean squared error (RMSE) and linear regression ($R^2$, slope and intercept). $Q_{10}$ values were selected on low bias (slopes and intercepts within 2 standard deviations of 1 and 0 respectively) due to the relative insensitivity of the $R^2$ and RMSE metrics (see Figure 1). By selecting the parameter based on model-data

fit instead of deriving a direct $Q_{10}$ value for each site and using the distribution, we demonstrate the robustness of the model and have a clear metric to select the parameter range. To test for statistical power, we randomly sampled the data 1000 times with sample sizes from 5 to 34 sites and compared this to samples with randomly assigned control vs warming (for each study

the percent carbon of control and treatment has a 50% chance of being switched). These random and sample generated $Q_{10}$ distributions, were compared using a two-sample Kolmogorov-Smirnov test to test that the distributions were statistical distinct.

### 2.4 Earth system model analysis

Earth system model simulations were drawn from CMIP5, the Coupled Model Intercomparison Project to support the 5th IPCC assessment report (Taylor et al., 2011). We downloaded simulation outputs from the RCP 8.5 scenario, representing the 'business as usual' scenario, including heterotrophic respiration (*rh*), soil temperature (*tsl*), and heterotrophic carbon stock (*cSoil* and *cLitter*,) from the CMIP5 repository on the Earth System Federation Grid. Ten-year means were taken at the beginning and end of the 21st century for each variable (corresponding to 2006-2015 and 2090-2099). Soil temperature was

averaged for the first 10 cm to correspond with experimental soil temperature readings. Soil carbon stock was calculated by adding all heterotrophic-respiring pools (including soil *cSoil* and litter *cLitter*) where multiple pools were reported. Soil carbon inputs were calculated from the monthly change in soil carbon stock plus the reported heterotrophic respiration. Model variable summaries can be found in Tables SI3 and processing code is documented in SI.

These 20 Earth system models are built from previous models which contain 10 distinct soil sub-models (Table 1). The number

of soil carbon pools in these ESMs varied from 1 (INM-CM4) to 8 (BCC-CSM1.1) with most models having 2 to 5 pools. None of the models reported soil carbon with depth, although GFDL documents a depth dependent model. There were three classes of pool structure for these models: *independent* where there was no exchange between soil carbon pools, *cascade* where pools with faster turnover times passed carbon to pools with slower turnover times, and fully *feedback* models where carbon was exchanged between faster and slower pools and vis-versa. In this set of models; 2 of these soil models were full feedback

models (HadGEM, ISPL-CM), 6 were cascade pool structure (MRI-ESM1, MIROC-ESM, MPI-ESM, CLM4.0 [CESM1, CCSM4, NorESM1], CanESM2, BCC-CSM1.1), and 2 were independent pools (GFDL-ESM2, INM-CM4). Only 2 model documented an explicit constant $Q_{10}$ (INM-CM2 and HadGEM2, $Q_{10}$=2), 1 model documented a soil temperature dependent $Q_{10}$ (CanESM2), 4 models documented a soil temperature sensitivity from Lloyd and Taylor which behaves very similar to $Q_{10}$=2 under moderate temperatures (Lloyd and Taylor, 1994), and the remaining 3 (ISPL-CM5, GFDL-ESM2, BCC-CM1.1)

all used a variation of the soil temperature sensitivity proposed in CENTURY (Parton et al., 1987, 1988) which also behaves very similar to $Q_{10}$=2 under moderate temperatures but declines at high temperatures (Lloyd and Taylor, 1994). The ESMs considered had a single global $Q_{10}$, or $Q_{10}$-formula dependent on soil temperature, uniformly applied to the decay pools. This documented structure should be approached with caution due to frequent lags between model development and documentation, actual values and functions may differ. For details with citations see Table 1.

Soil carbon stocks at the beginning of Earth system model simulations are typically documented to be spun up to close to steady state, and there is numerical support that this holds throughout the simulation (see Results and Figure S3). Thus Eq. 3 can be extended to the change in soil carbon stock over the 21st century. This leads to the following explicit calculation for a $Q_{10}$ value at each grid cell.

$$\ln(Q_{10}) = \left(\frac{10}{T_m - T_f}\right) \ln\left(\frac{C_f\, u_m}{C_m\, u_f}\right), \tag{5}$$

where the $Q_{10}$ value is related to the modern soil temperature $T_m$, future soil temperature at the end of the 21st century $T_f$, modern soil inputs $u_m$, future soil inputs $u_f$, modern soil carbon stock $C_m$ and the future soil carbon stock $C_f$.

For soils that are very close to zero soil carbon stocks, have minimal shifts in soil temperature or have very low soil inputs, the estimated $Q_{10}$ is not finite. Similarly soils which are not well described by their shift in soil temperature (for example, if there is a significant shift in the moisture regime) may have non-typical $Q_{10}$ values that are either less than 0.5 or greater than 5. We examined the amount of shift in soil carbon stocks associated with the four categories of $Q_{10}$ values (nonfinite, less than 0.5, greater than 5, or typical), as well as the spatial patterns associated with these categories.

To support the assertion that the $Q_{10}$ value can be calculated from relatively short time scales found in the field experiments, we examined the distribution typical $Q_{10}$ values associated with similar soil temperature steps experienced by the field experiments at 1, 5, 10, 15, 20, 50, 75, and 84 year time-scales using 10-year mean gridded values of soil carbon stocks, soil inputs, and soil temperature. It should be noted however that changes in the moisture conditions over the 21st century may complicate this analysis of the Earth system model simulations, thus it is not an exact proxy for the field experiments where the control and treatment plots experienced similar baseline climate conditions and a more or less constant offset throughout the experiment.

Finally, the $Q_{10}$ distribution was scaled to reflect the best estimate and uncertainty from the field data. This distribution shift was done by normalizing the $Q_{10}$ map to the mean of the distribution and multiplying it by the experimentally derived values. The $Q_{10}$ correction was only applied to grids with typical $Q_{10}$'s (non-typical $Q_{10}$'s were considered to have predominately non-temperature driving variables and their soil carbon stocks were not altered). This normalization shifted the global $Q_{10}$ distribution within the models to match the most common (geographically likely) $Q_{10}$ with the data-driven $Q_{10}$ value, yet by preserving the distribution we preserved other factors affecting changes in decomposition rate (i. e. moisture shifts) in the model. We then recalculated the change in soil carbon for each grid cell with this modified $Q_{10}$ according to Eq. 3 and calculated the global area-weighted totals.

The full analysis script and those used to generate the figures are available in the supplemental.

**3 Results**

From the changes in soil carbon stocks across field studies, we find a global $Q_{10}$ of 2.2 (90%CI 1.6, 2.7; $R^2 > 0.95$, root mean squared error $< 2$; Figure 1, Figure S2). The model-data fit was evaluated using a linear regression and root mean squared error (Figure 1). While the $R^2$ of the model-data comparison was relatively insensitive to the $Q_{10}$ value, there was a notable improvement in the bias with $Q_{10}$ (as defined as the slope within 2 standard deviations of 1 and intercept within two standard deviations of 0). This bias-driven criteria was used to select $Q_{10}$ values from a prior range of (0.1, 5), see Methods for details.

The $Q_{10}$ distribution was compared with a random null distribution and was significantly distinct (Kolmogorov-Smirnov D=0.441, p < 2e-16, See Table S2, Figures 2 and S1). Randomly selecting 5 to 34 sites from the full dataset were compared to a null distribution where control vs warmed labels were randomized. The quartiles of the data subsets notably converged at a sample size of 25 where the null distribution was relatively invariant across sample size (Figure 2). The distribution of the $Q_{10}$

values under null appeared log-normal, centered around 1 demonstrating no temperature effect (Figure S1). The distribution of the $Q_{10}$ range for the data subsets converged to around 2.2 (Figure S1).

The balance between gridded soil inputs and heterotrophic respiration at both the initial and final 10-year mean for the 21st century was within 10% for over 93% of the grid cells across all models with half of the grid cells within 0.1%. Most models had 95% and two models consistently had 100% of their grid cells within 10% -- the absolute value of the net flux was within

10% of the highest primary flux (Figure S3). Thus, the soil inputs are on the same order of magnitude as the soil outputs. This was reflected in very similar $Q_{10}$ distributions regardless of whether soil inputs or heterotrophic respiration was used to derive the $Q_{10}$ value (Figure S4). A notable exception to this was the MIROC-ESM model which did see differences in inputs and heterotrophic respiration drive different $Q_{10}$ distributions (Figure S4).

The inferred $Q_{10}$ values in the Earth system models derived from 10-year mean changes across different time steps (1, 5, 10,

15, 20, 50, 75, and 84 years) had similar distributions in most of the models (Figure S4). There were minor shifts in the mode of most models which could be attributable to changes in the moisture conditions or other (non-temperature or input) environmental variables in the simulation. Models aggregated across common land models showed marked similarity in their $Q_{10}$ distributions (Figure S5). There was also an extremely high correlation between $Q_{10}$ values derived from soil inputs compared to those derived for heterotrophic respiration across all models (Figure S6).

The inferred $Q_{10}$ values in the Earth system models from the decadal average across the 21st century fell into four categories (nonfinite, less than 0.5, greater than 5, or typical; Figure S7), however most of the change in soil carbon stocks over the 21st century occurred in grid cells with typical $Q_{10}$ values between 0.5 and 5 (Figure S6). A notable exception to this trend was the MRI-ESM1 model where roughly half of the change in carbon stocks occurred in grid cells with $Q_{10}$ values greater than 5 (Figure S6). Spatially the $Q_{10}$ categories showed strong geographical patterns (Figure S7). The GFDL-ESM2 models were

dominated by non-finite values in high northern latitudes (Figure S7). MIROC-ESM, CCSM4, CESM1, and NorESM1 models were dominated by $Q_{10}$ values above 5 in the high northern latitudes (Figure S7). Unless otherwise noted, only typical $Q_{10}$ values are addressed for the remainder of this study.

The inferred $Q_{10}$ values for the decadal average across the 21st century, also showed strong geographic patterns (Figures 3) and was typically unimodal (Figure S6). MIROC-ESM and MIROC-ESM-CHEM showed the weakest spatial patterns with

high grid-to-grid variation (Figure 3). Mean $Q_{10}$ values fell within the 90% CI of the field data $Q_{10}$, ranging between 1.8 (CESM1(CAM5), HadGEM2-ES, ISPL-CM5A, and MPI-ESM-MR) and 2.6 (MIROC-ESM-CHEM), with the multi-center $Q_{10}$ values at $2.0 \pm 0.2$ (Tables 2).

When the inferred $Q_{10}$ values were modified to reflect the data-driven $Q_{10}$ range, resulting variation in the multi-center mean was almost as large as the variation across model projections (Figure 4, Table 2). Re-centering the global $Q_{10}$ distribution to

reflect the range of field-driven $Q_{10}$ values (Figure S8) resulted in changes in soil carbon stocks over the 21st century of between -452 Pg-carbon (MPI-ESM-MR) and 525 Pg-carbon (HadGEM2-CC) with a best-estimate $Q_{10}$ ($Q_{10} = 2.2$) resulting in $19 \pm 155$ Pg-carbon (multi-center mean $\pm$ 1 SD) and field-drive bound ($Q_{10} = 1.6, 2.7$) of $[248 \pm 191, -95 \pm 209]$ Pg-carbon (Figure 4, Table 2).

**4 Discussion**

By capturing information about warming induced changes to relatively undisturbed field soil carbon stocks directly rather than inferring this from soil respiration rates, this is the first study to generate field $Q_{10}$ estimates of soil carbon losses without needing to correct for belowground autotrophic respiration. Using a simplified version of a traditional decomposition model with a soil temperature sensitivity function, we estimate that the global $Q_{10}$ value is 2.2 ([1.6, 2.7] 95%CI, Figure 1, S2). This

$Q_{10}$ is notably higher than previous global estimates based on field soil respiration data ($Q_{10} = 1.4$ to 1.5 (Bond-Lamberty and Thomson, 2010; Mahecha et al., 2010)), yet well within the range of estimates from laboratory-based studies (Davidson and Janssens, 2006) as well as close to documented soil temperature sensitivity parameters (~2) of Earth system models (Table 1). This $Q_{10}$ range is statistically significant. Resampling the 36-study data set demonstrates the need for over 25 sites to distinguish the $Q_{10}$ range from random (Figure 2 and S1). While the $Q_{10}$ distribution for the 34-study subset is distinct from

the null (Kolmogorov-Smirnov D=0.441, p < 2e-16), there appears to be some minor drift in the range suggesting that more study sites could be informative and we hope future studies will include data recently identified (van Gestel et al., 2018). Inferring a decadal-scale environmental sensitive from an annual-scale experiment is generally controversial. However, in this case, traditional model structures assume a temperature sensitivity function that is invariant across space and time and numerical trends in the Earth system model reflect this. In the traditional model structure the soil temperature sensitivity

function is applied as a single scaler to multi-pool models causing the relative decomposition response in both fast and slow pools to be the same (for example, (Parton et al., 1987)). Examining the inferred gridded $Q_{10}$ values from annual means across time scales from 1 year to 84 years in Earth system models shows a strong similarity in the distribution most models (Figure S4). Similarly using soil inputs as opposed to heterotrophic respiration did not affect the distribution of the gridded $Q_{10}$ values, with the notable exception of MIROC-ESM which is explained by unusual differences in soil inputs and outputs (Figure S3,

S4). Differences in the $Q_{10}$ distribution across time scales are likely driven then by interaction with other sensitivity functions like moisture or in shifts in the allocation of dead vegetation to different pools as the plant type distribution changes over time. If soils are more sensitive to warming than previously expected, then how would this affect future soil carbon stocks over the 21st century? To address this question, we turned to the CMIP5 Earth system models run under RCP 8.5 (Taylor *et al.*, 2011). In order to modify the Earth system model output to reflect the data-driven $Q_{10}$ we applied similar assumptions used in the

field-data analysis. We first examine the soil temperature sensitivity of CMIP5 Earth system model simulated soil carbon stocks. In contrast to the field data, we take into account the effect of the change in soil inputs on soil carbon stocks in the Earth system models because these coupled simulations include $CO_2$ fertilization and other climate effects known to influence

primary production (see Methods, Eq. 5). Though these inferred $Q_{10}$ values ($Q_{10}$ = 1.8, 2.6) fall within the uncertainty of the field derived $Q_{10}$ values ($Q_{10}$ = 1.6, 2.7), most ESM-$Q_{10}$ means fell under the median data-$Q_{10}$ value of 2.2 (Table 2) implying ESMs were, on average, less sensitive to soil temperature shifts than the field warmed data would imply. It should be noted that this inferred $Q_{10}$ value is not exactly the parameterized $Q_{10}$ value, and is instead a combination of the soil temperature

sensitivity and other environmental sensitivities. If there were, on average, an additional constraint on respiration (such as moisture) we might expect the inferred $Q_{10}$ parameter to be lower than the model parameterized $Q_{10}$.

There were notable regional patterns across all but 2 of the Earth system model inferred $Q_{10}$ (Figure 3, S7). High northern latitudes tended to have either large or non-finite $Q_{10}$ values suggesting that something other than soil temperature and input shifts were driving changes in soil carbon stock. This alternative driver could be a shift in moisture regimes or dynamics driven

by thaw thresholds which could similarly affect the analysis of the field-data. Additional drivers of soil decomposition dynamics, beyond temperature and inputs considered here, have the potential for explaining some of the variation in the $Q_{10}$ range and new model structures are being explored to take some of these mechanisms into account (Luo et al., 2015; Wieder et al., 2015a). This remains an active area of research.

Propagating this field-$Q_{10}$ range into the ESM projections resulted in greater carbon losses from the soil by the end of the 21$^{st}$

century (multi-center means the of soil carbon change, shifted from 88 to 19 Pg-carbon) with large uncertainties; ESM multi-center standard deviation was initially 152 Pg-carbon which is half of the range in multi-model mean attributed to $Q_{10}$ 95% CI [248, -95] Pg-carbon (Figure 4, Table 2). To calculate these modified projections, means of the model specific $Q_{10}$ distributions were re-centered to reflect the best estimate $Q_{10}$ and associated 95% CI from the field data analysis. By preserving the distribution within the model, we attempted to propagate soil moisture sensitivities and other model-specific effects into the

modified projections. We also did not modify grid cells with non-typical $Q_{10}$ values (non-finite, below 0.5 or above 5) since those grids likely governed other non-temperature drivers. The large range of carbon shifts in each ESMs driven by this $Q_{10}$ CI confirms the importance of considering parameter uncertainty in the land carbon component of Earth system model projections. The post-hoc correction that we present provides an innovative way to account for this parameter variation without the computational burden of additional ensemble runs.

This analysis includes several basic assumptions and caveats. Specifically, we assume that the difference between treatment and control is driven entirely by the soil warming effect, and those warming effects are uniform across soil carbon quality. Though warming-induced changes in soil inputs are, on average, relatively small, they are have been shown to be highly variable in similar sites (Lu et al., 2013). The analysis of field data could be extended to account for these changes in inputs in follow-up studies (Eq. 3). A large increase in soil inputs would cause an underestimation of the $Q_{10}$ value, while a decrease in

soils inputs would cause an overestimation of the $Q_{10}$ value (see Eq. 3). While there is some evidence to support soil temperature sensitivity dependency on soil carbon quality (Knorr et al., 2005), there is also evidence for a uniform soil temperature sensitivity (Hicks Pries et al., 2017). as is represented in the Earth system models considered in this study (Table 1). A quality dependent $Q_{10}$ would not be separable from the bulk decay term and thus a one pool model would be inappropriate

in this case (see SI). In addition, the dataset has acknowledged biases (see Crowther *et al.*, 2016), which are typical of field studies.

### 4.1 One pool simplification

We find that multi-year soil carbon dynamics can be well-described by a one pool model at a specific time scale in both the
Earth system models and field experiment. If we restrict the decomposition models to those with either independent or cascade pool structures (that is no carbon passed from the slow to the fast pools) then the temporal dynamics of the total soil carbon of the system at a specific time scale can be approximated by a single pool due to the fact that the lower triangular decomposition-transfer matrix is diagonalizable (see SI for details). While this diagonalizable property does not hold for full feedback models where carbon is transferred from the slower to faster carbon pools, all decomposition-transfer matrices are M-matrices. If we
combine the positive-inverse properties resulting from this M-matrix structure and assume that the soils are close to metastable state (that is soil inputs are roughly equal to the heterotrophic respiration outputs, as we show in Figure S3 for the Earth system models considered and would expect for soils from intact systems). Then the total soil carbon can be described by a bulk decay rate that is a linear combination of the transfer coefficients, decay rates, and input allocations of the component pools (see SI for analytical details). This provides analytical support for the one pool simplification seen numerically in the Earth system
models in the CMIP5 project (Todd-Brown *et al.*, 2013, 2014).

The one pool simplifications described above are controversial assertions. The one pool model has proven inadequate to describe laboratory incubations where heterotrophic respiration over time is compared to the soil carbon stock (Thornton, 1998; Weng and Luo, 2011). This is due to the multiple time scales considered (daily, monthly and annual) and, more importantly, the fact that these laboratory incubations are by their nature not at steady state since any inputs to the system are
generally removed. Thus this analysis would not be expected to hold for laboratory incubation and we would further expect the bulk decay rate to change with time scales for sites undergoing rapid changes in inputs (in other words, the bulk decay rate inferred at a 1 year time step would not match the 100 year time step at a site undergoing transition from grassland to forest). Another key assumption is that soil organic carbon of different quality response the same to warming. However, the scalar multiplier representing environmental sensitivities are independent of pools in most models (ex (Parton et al., 1988)). These
scalar multiplies (like the $Q_{10}$ temperature sensitivity examined in this study) would be invariant to time scale if this modeling assumption is applied to the field analysis. Finally shifts in the allocation of dead vegetation to the different soil pools would shift the bulk decay rate of the one pool approximation (see SI: Analytical proofs). With these caveats in mind, we feel that the one pool approximation is extremely valuable in analyzing soil carbon models and data.

### 5 Conclusion

It is still unclear how the terrestrial carbon cycle in general, and soils in particular, will respond to climate change over the 21$^{st}$ century. The CMIP5 models, representing our best coupled climate models to date, have a wide range of soil carbon responses

over the 21$^{st}$ century (Todd-Brown et al., 2014). While it would be nice to have all the models agree on a tightly bound answer, the question we should be asking scientifically is: Does the variation in the models reflect our best scientific understanding? Models must capture not only mean trends but also system variance and accurately represent scientific uncertainty.

Post-hoc correction of simulation results can provide some insight into known gaps in Earth system models without the computational hurdle of re-running simulation results. Previous studies have applied post-hoc corrections to address nutrient limitations on net primary production (Wieder et al., 2015b) and this study demonstrates the high level of uncertainty that can be driven by the soil temperature response parameter. This study suggests that soil carbon response to warming is highly variable in the field and ESMs should increase their variability to reflect this field-variation. Future studies increasing the number of field-warmed studies (van Gestel et al., 2018), as well as extending the field data to include changes in plant productivity in response to warming would inform the field-derived $Q_{10}$ analysis explored here. In addition, explaining field moisture and applying that understanding to a post-hoc Earth system model analysis is a logical next step.

## 6 Acknowledgements

We thank Drs Ben Bond-Lamberty and Vanessa Bailey for their reviews of this manuscript before submission, Drs Will Wieder and Mark Bradford for discussions in formulating this analysis, and Donald Jorgensen of for his work on the graphical abstract. We would also like to thank the reviewers for this manuscript for their feedback.

Part of the research described in this paper was conducted under the Laboratory Directed Research and Development Program at Pacific Northwest National Laboratory, a multiprogram national laboratory operated by Battelle for the U.S. Department of Energy; KT-B is grateful for the support of the Linus Pauling Distinguished Postdoctoral Fellowship program. TWC was supported by a Marie Skłodowska Curie fellowship. BZ was supported in part by the U.S. Department of Energy, Office of Science, Office of Advanced Scientific Computing Research as part of the Uncertainty Quantification for Complex Systems project.

We thank the experimentalists who generously made their data available for this analysis.

We acknowledge the World Climate Research Programme's Working Group on Coupled Modelling, which is responsible for CMIP, and we thank the climate modeling groups (listed in Table SI3) for producing and making available their model output. For CMIP the U.S. Department of Energy's Program for Climate Model Diagnosis and Intercomparison provides coordinating support and led development of software infrastructure in partnership with the Global Organization for Earth System Science Portals.

The authors have no conflict of interest to declare.

## 7 Author Contributions

KTB and TWC both contributed to the writing of this manuscript. KTB was responsible for the analysis. TWC coordinated the field data set. BZ was responsible for the mathematical analysis and made contributions to the revisions of the manuscript.

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

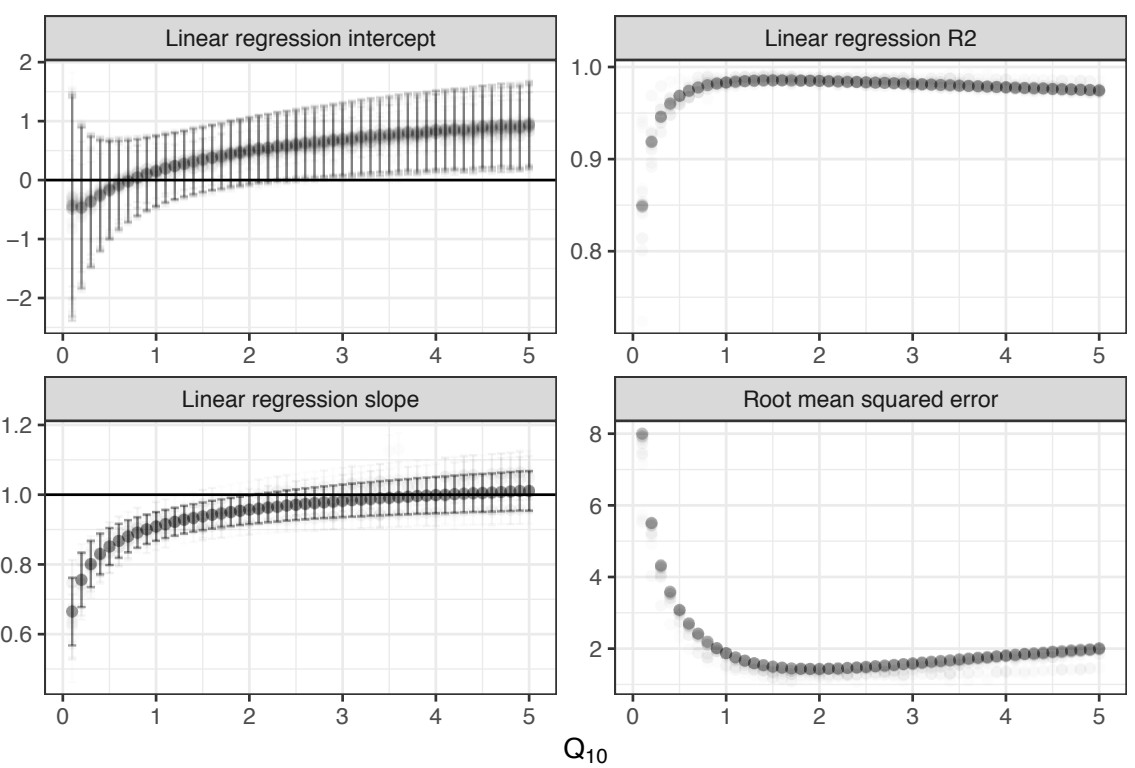

**Figure 1: The model-data fits across different $Q_{10}$ values for random subsets of 34 sites including the root mean squared error, and linear regression metrics $R^2$, slope, and intercept. The model is take from Eq. 4 ($C_w = C_c Q_{10}^{-\Delta T/10}$). Slope and intercept values are**
15    **shown with 2 standard deviation error bars.**

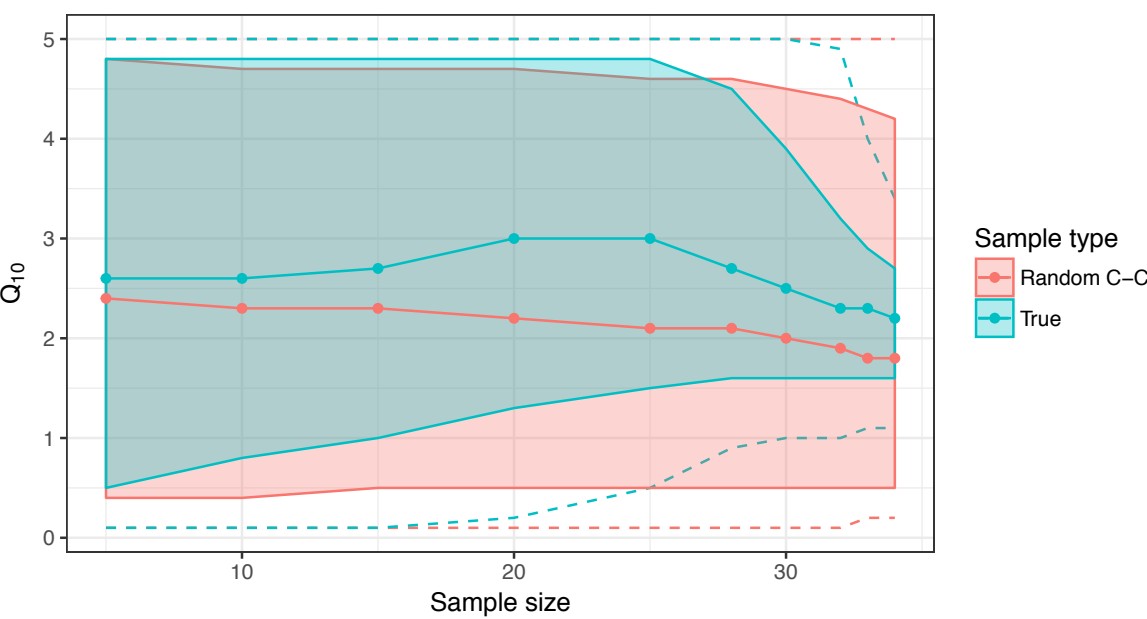

**Figure 2: The $Q_{10}$ with good 1-to-1 model-data fits defined in Figure 1, at 90% confidence interval (band) with minimum and maximum values (dotted line) and median value (solid line), across 10 different sample sizes ranging from 5 to 34, for the original data set (True: blue) and randomized case-control (Random C-C: red).**

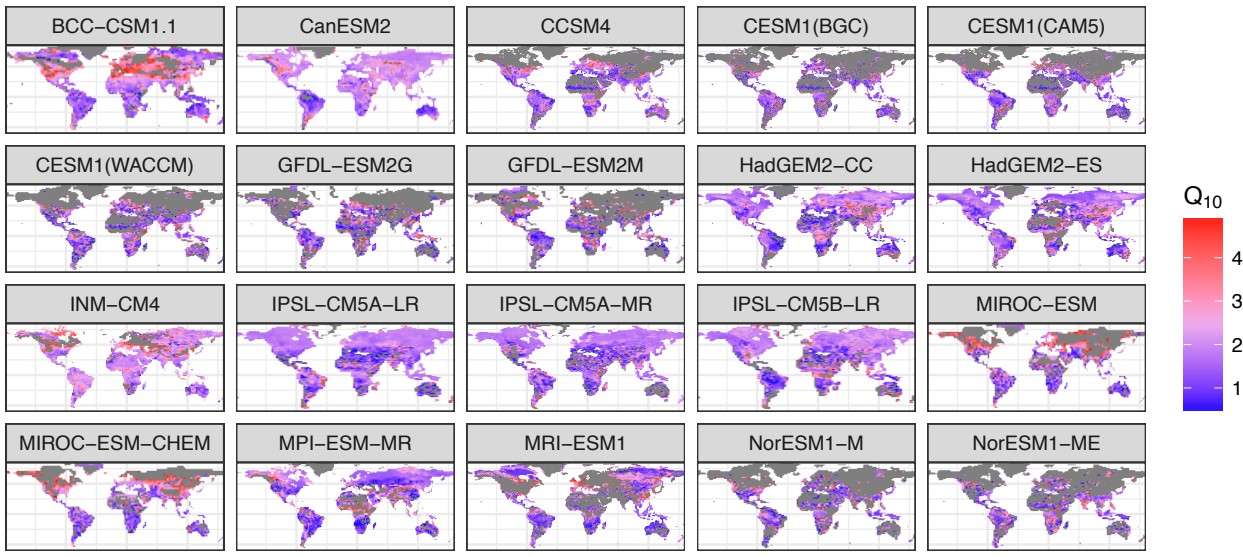

**Figure 3: Inferred $Q_{10}$ values from the Earth system models (CMIP5, RCP 8.5). The colour scheme is centered around the field-driven $Q_{10}$ median value of 2.2. Grey indicates non-typical $Q_{10}$ values that were either non-finite, less than 0.5 or greater then 5.**

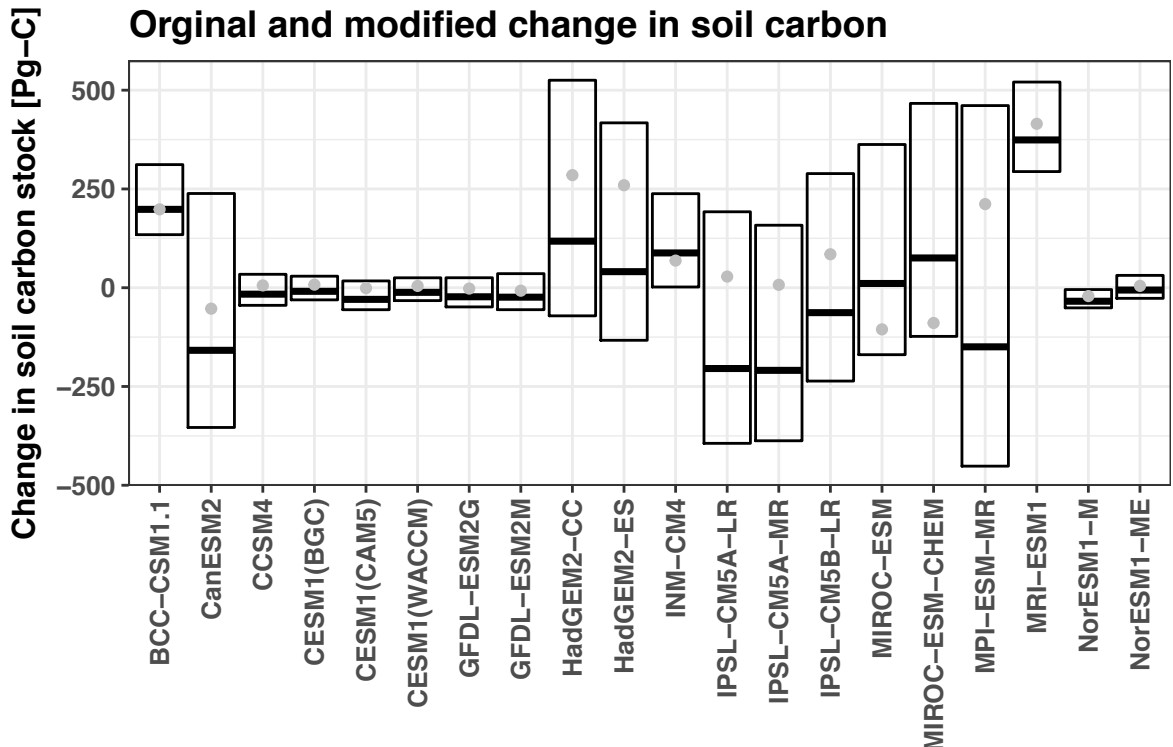

**Figure 4: Changes in soil carbon stock (10 year means) over the 21st century from Earth system models (RCP 8.5). Grey dots are the original estimates, the open box is the soil carbon loss after the $Q_{10}$ is rescaled using the 2.5%, 50%, and 97.5% quartiles from the field data**

| Model Center | Earth system model | Soil/land carbon sub-model | Number of soil carbon pools | Pool structure | Temperature sensitivity | Citations |
|---|---|---|---|---|---|---|
| BCC | BCC-CSM1.1 | BCC_AVIM1.0; AVIM2; CEVSA | 8 | Cascade | f(T) | (Cao and Woodward, 1998; Ji et al., 2008; Wu et al., 2013) |
| CCCma | CanESM2 | CTEM | 2 | Cascade | $Q_{10}(T)$ | (Arora, 2003; Arora et al., 2011; Arora and Boer, 2010) |
| NCAR | CCSM4 | CLM 4.0; CN | 7 | Cascade | Lloyd and Taylor | (Gent et al., 2011; Lawrence et al., 2011; Oleson et al., 2010; Thornton et al., 2007; Thornton and Rosenbloom, 2005; Tjiputra et al., 2013) |
| NSF-DOE-NCAR | CESM1(BGC) | | | | | |
| | CESM1(CAM5) | | | | | |
| | CESM1(WACCM) | | | | | |
| NCC | NorESM1-M | | | | | |
| | NorESM1-ME | | | | | |
| NOAA GFDL | GFDL-ESM2G | LM3.0; ED | 2 | Independent | f(T) | (Dunne et al., 2013; Moorcroft et al., 2001; Shevliakova et al., 2009) |
| | GFDL-ESM2M | | | | | |
| MOHC* | HadGEM2-CC | ROTHC | 4 | Feedback | $Q_{10}=2$ | (Coleman and Jenkinson, 1999; Collins et al., 2011) |
| | HadGEM2-ES | | | | | |
| INM | INM-CM4 | LSM 1.0 | 1 | Independent | $Q_{10}=2$ | (Bonan, 1996; Volodin, 2007) |
| IPSL | IPSL-CM5A-LR | ORCHIDEE | 5 | Feedback | f(T) | (Dufresne et al., 2013; Krinner et al., 2005) |
| | IPSL-CM5A-MR | | | | | |
| | IPSL-CM5B-LR | | | | | |
| MIROC | MIROC-ESM | SEIB-DGVM | 3 | Cascade | Lloyd and Taylor | (Sato et al., 2007; Watanabe et al., 2011) |
| | MIROC-ESM-CHEM | | | | | |
| MPI-M | MPI-ESM-MR | JSBACH | 5 | Cascade | Lloyd and Taylor | (Giorgetta et al., 2013; Schneck et al., 2013) |
| MRI | MRI-ESM1 | LPJ-DGBM | 3 | Cascade | Lloyd and Taylor | (Adachi et al., 2013; Obata and Shibata, 2012; Sitch et al., 2003) |

Table 1: This is a summary of the soil decomposition sub-models for the ESMs used in this study and includes the number of pools, structure of the carbon exchange between those pools, temperature sensitivity function, and citations. Temperature sensitivity

function is either denoted as: **f(T): borrowed from the Century model** (Parton et al., 1987, 1988)**, Lloyd and Taylor: taken from the recommended form from** (Lloyd and Taylor, 1994)**, $Q_{10}(T)$: a temperature dependent $Q_{10}$ as defined by (Arora, 2003), or a $Q_{10}$ parameter for the $Q_{10}$ function as defined in this manuscript.**

| | SOC | Rel. | dT | | dSOC | dSOC | dSOC | dSOC |
|---|---|---|---|---|---|---|---|---|
| | [Pg-C] | Inputs | [°C] | $Q_{10}$ | [Pg-C] | $Q_{10}$=1.6 | $Q_{10}$=2.2 | $Q_{10}$=2.7 |
| BCC-CSM1.1 | 1050 | 1.40 | 3.7 | 2.2 | 198 | 312 | 198 | 134 |
| CanESM2 | 1541 | 1.29 | 7.1 | 2.0 | -53 | 239 | -158 | -354 |
| CCSM4 | 515 | 1.32 | 4.2 | 1.9 | 6 | 34 | -16 | -45 |
| CESM1(BGC) | 515 | 1.29 | 3.8 | 1.9 | 8 | 29 | -9 | -31 |
| CESM1(CAM5) | 553 | 1.30 | 4.6 | 1.8 | -1 | 17 | -30 | -56 |
| CESM1(WACCM) | 502 | 1.32 | 3.9 | 1.9 | 5 | 25 | -12 | -33 |
| GFDL-ESM2G | 1422 | 1.41 | 5.1 | 1.9 | -2 | 25 | -23 | -49 |
| GFDL-ESM2M | 1278 | 1.38 | 4.5 | 2.0 | -8 | 36 | -24 | -56 |
| HadGEM2-CC | 1122 | 1.55 | 8.4 | 1.9 | 285 | 525 | 118 | -71 |
| HadGEM2-ES | 1129 | 1.56 | 8.3 | 1.8 | 259 | 417 | 41 | -133 |
| INM-CM4 | 1688 | 1.27 | 3.3 | 2.3 | 69 | 238 | 88 | 2 |
| IPSL-CM5A-LR | 1361 | 1.48 | 8.2 | 1.8 | 28 | 192 | -205 | -394 |
| IPSL-CM5A-MR | 1403 | 1.43 | 7.6 | 1.8 | 7 | 158 | -209 | -387 |
| IPSL-CM5B-LR | 1274 | 1.41 | 7.6 | 1.9 | 85 | 289 | -63 | -236 |
| MIROC-ESM | 2586 | 1.35 | 7.2 | 2.5 | -105 | 363 | 11 | -170 |
| MIROC-ESM-CHEM | 2588 | 1.30 | 7.3 | 2.6 | -89 | 467 | 75 | -123 |
| MPI-ESM-MR | 3110 | 1.31 | 6.3 | 1.8 | 212 | 461 | -150 | -452 |
| MRI-ESM1 | 1452 | 1.52 | 4.4 | 2.0 | 415 | 521 | 374 | 294 |
| NorESM1-M | 547 | 1.31 | 3.7 | 1.9 | -21 | -4 | -34 | -51 |
| NorESM1-ME | 553 | 1.32 | 3.6 | 2.0 | 5 | 31 | -6 | -27 |
| Multi-center mean | 1403 | 1.37 | 5.4 | 2.0 | 88 | 248 | 19 | -95 |
| Multi-center sd | 793 | 0.09 | 1.8 | 0.2 | 153 | 191 | 155 | 209 |

**Table 2**: **Global model summary with multi-center mean and standard deviation for modern soil organic carbon (SOC) stocks [Pg-C], relative shift in soil inputs ($\frac{u_f}{u_m}$), absolute change in soil temperature (dT) [°C], inferred mean of $Q_{10}$ as calculated by grid cell (see Eqn 5), the change in soil organic carbon (dSOC) over the 21st century [Pg-C], and the change in soil organic carbon with rescaled $Q_{10}$ values (1.6, 2.2, and 2.7).**