# Peer review of "Field-warmed soil carbon changes imply high 21st century modeled uncertainty"

_Biogeosciences, 2018_

## Referee Comment (RC1) · C.D. Jones (Referee) · 2 Mar 2018

Review of "Field-warmed soil carbon changes imply high 21st century model uncertainty", by K Todd-Brown et al.

This is an interesting and well written paper addressing a subject area of clear importance and relevance to the readership of Biogeosciences.

The authors clearly articulate the issue of uncertainty in future land-carbon storage, how this responds to climate change and the vital role played of soil organic carbon and its sensitivity to temperature. The results lead to a change in a central estimate, across multi-model results, of soil carbon changes by 2100, although interestingly the data constraint does not reduce the overall uncertainty. This highlights the claim made in

the title of "high uncertainty" and leave this area as an outstanding issue for modellers and process community alike to address.

I recommend publication after addressing a few minor concerns as below.

Chris Jones

The paper continues the work and analysis of these authors from the Crowther et al. 2016 Nature paper. The dataset assembled of soil warming plots and changes in soil carbon is clearly very valuable for understanding this key issue of Earth System Modelling. I much prefer the approach here, by comparing with process-based models and trying to constrain their simulations, than I do the direct extrapolation to global scale as per Crowther et al. So overall I like the approach deployed here and have mainly minor comments. There are clearly assumptions and choices made which affect the results - I don't believe these should prevent publication - the authors appear to have been careful to consider these and discuss their implications.

2 specific examples are:

- the issue of timescales - the approach to diagnose Q10 relies on pseudo-equilibrium of the system. It could therefore be potentially dangerous and misleading to compare the outcomes from a Q10 diagnostic of 2-5 years warming with ESM studies of 100 years. I was pleased to see this tested explicitly and the SI shows detailed reasons to believe that this is not a large confounding issue.

- the issue of using a single-pool simple model to recreate and re-scale the responses. Again, the authors appear to have taken potential concerns seriously and present good arguments why their approach is OK. I suspect that using a two-pool model would give different answers (inevitable for modelling!) - but it's not clear that this simplification is inappropriate. Given that prominent ESMs in the past, and still some at CMIP5, still deploy a single soil carbon pool it is reasonable for a simple-model approach to do so (by definition simple models bring clarity at the expense of detail - I feel it is well argued

here that the choice is appropriate).

I have one issue with some of your methodological choices

- you sample q10 values down to 0.1, and in your ESM scaling define "typical" as going down to q10=0.5. It seems VERY odd (unphysical even) to take q10 below 1. Given this is not a linear relationship, you are not looking at taking gradients down towards 0 which represent being flat. Rather q10=1 is where you hit "flat" - i.e. no sensitivity at all to temperature. Once you go below q10=1 then you get an inverse relationship which says that respiration increase as you get COLDER. I'm not aware of any way this could be possible. So I would strongly suggest you take q10=1 as your lower boundary.

specific minor comments:

- please be very careful discussing response to "temperature" - it is always important to specify clearly if this means air- or soil-temperature. It has long been known that the apparent q10 is much lower when calculated as a response to air temp (Raich & Potter (1995, GBC) suggest a Q10 value of 2.0 for soil temperature is roughly equivalent to 1.7 for air temperature). Please check especially the Bond-Lamberty paper/comparison - they calculate a low Q10, as you say - but isn't this for air-temp? The comparison is then perhaps misleading

- be careful using cCwd - is this a distinct carbon pool? or is it a sub-component of cLitter? I believe that the total vegetation system is captured by cVeg+cLitter+cSoil (e.g. see Jones et al., 2013, J.Clim). Then, within this, cLitter is split into tier-2 variables of cCwd, cLitterAbove, cLitterBelow - these are intended to allow reporting in greater detail, but are not new pools. So I think you should remove any cCwd data from your study. I appreciate this is not well explained in the CMIP5 data request - we tried to clarify this for CMIP6 - see figure 5 of Jones et al (C4MIP documentation paper, GMD, 2016)

- Although HadGEM2-ES does base it's soil carbon scheme on RothC it is not identical

- in particular we chose to keep a uniform q10=2 rather than the RothC temperature function (which we found to be not well behaved at very low T). Otherwise your description of HadGEM2 is correct.

- the discussion correctly discusses the possible role of soil moisture as a rate modifier. Also to consider are vegetation cover, soil quality, soil structure and changes in input quality. Some models, like RothC, change their decomposition according to overlying vegetation. They also change their allocation (your "b" matrix in equation 1) according to vegetation type and lability of litter inputs. I don't think these are major factors, but worth mentioning its not just T and moisture which change the respiration.

---

## Referee Comment (RC2) · Anonymous Referee #2 · 12 Mar 2018

This is a great study, in which Todd-Brown et al. creatively combined experimental soil warming data with Earth system model outputs to illustrate the observation informed uncertainty in future responses of soil organic carbon to warming. I'd recommend publication after addressing a few issues outlined below.

1. While it's hard to move away from steady state assumption (it leads to very convenient mathematical forms), I think it is important to at least illustrate the bias that steady state assumption can have on estimated $Q10$. It would be hard to do for observed SOC responses to warming without very crude assumptions about base turnover rate, but seems realistic for the ESM output. The term on the left side of equation 1 can be extracted from the ESM output and used instead of 0, and baseline k as well as $Q10$'s can be reverse-engineered. I think the steady state bias could be quite substantial,

because the 10% (or even 1%) difference between C input and output can have a cumulative effect over time. With observation-derived Q10 estimates, I suggest excluding sites that are with high degree of certainty are not in steady state. An example is Delta Junction site in Alaska, which experienced stand replacing fire in 1999. Even though warming experiment lasted for 10 years it takes many decades for the SOC to recover after a stand replacing fire (Fu et al., 2017), so equation 4 would not be appropriate for this site.

2. This leads me to equation 4. I would strongly suggest not omitting the ratio of control inputs to warmed inputs. As authors rightly pointed out, if ratio is less than 1, Q10's are going to be underestimated, and if warming leads to decrease in NPP, the Q10's would be overestimated. A meta-analysis study by Wu et al. (2011) illustrated that warming increases total NPP on average by 15%, and belowground NPP by 52%, which has very substantial implications for Q10 (please see the attached image illustrating the effect of change in productivity on Q10's for Delta Junction, AK). I think looking up changes in NPP estimates for every site would make the observation-based Q10 estimates more defendable.

3. Lastly it wasn't completely clear to me why authors chose to estimate observation-based Q10's with the method described in section 2.3 instead of directly solving for it using equation 3 or 4. Could you please touch on that in the updated version of the manuscript?

Below, please see a list of minor issues:

P1L25: "stimulate"?

P1L27: "(the opposing carbon flux)" can be omitted in my view

P3L20: It seems that the model is not accurately specified: vector C(t) of size n by 1 cannot be multiplied by n by n matrix K. The accurate specification of the negative term would be "A*(Q10*K)*C(t)"

P3L27: "scalar"?

P4L2: "vice-versa"?

P6L16: "data-driven"?

P6L18: "grid cell"

P8L10-11: Please revise the topic sentence, it's not very clear.

P8L16: did you mean Figure S3?

P8L25: typo: "ESM-Q10"
* * *
[Figure]

[Figure]

**Fig. 1.**

---

## Referee Comment (RC3) · Anonymous Referee #2 · 12 Mar 2018

P.S. here are the references used in the review:

Fu Z, Li D, Hararuk O, Schwalm C, Luo Y, Yan L, Niu S (2017) Recovery time and state change of terrestrial carbon cycle after disturbance. Environmental Research Letters, 12, 104004.

Wu Z, Dijkstra P, Koch GW, Peñuelas J, Hungate BA (2011) Responses of terrestrial ecosystems to temperature and precipitation change: a meta-analysis of experimental manipulation. Global Change Biology, 17, 927-942.

---

## Short Comment (SC1) · 12 Mar 2018

This is a well-written and original paper with findings that are relevant to the Biogeosciences community. The code documentation is particularly great.

I have reviewed for an earlier version of this manuscript, and they have successfully addressed my previous comments, adding some discussion of how a systematic change in input rates (which I understand is not in the dataset but is nevertheless likely) would affect carbon stocks and Q10, and justifying the use of a one-pool model. From the other referee comments, these points may warrant some additional discussion. Therefore, I recommend publication after minor revision.

Two additional thoughts: Most versions of the Century soil model embedded in a num-

ber of the ESMs listed as 'cascade' models (e.g., the CLM family models) are feedback models where some soil C is transferred from the slow and passive to the active pool. These versions of Century may well be simplified to the 'cascade' model type, but that development is not clear from the descriptions of the ESM soil models.

I know it's hot of the press, but because the authors base their analysis off of the Crowther et al. 2016 dataset, they could also mention the potential implications of the expansion on their analysis in the Van Gestel et al. comment.

van Gestel, N., Shi, Z., van Groenigen, K. J., Osenberg, C. W., Andresen, L. C., Dukes, J. S., ... & Reich, P. B. (2018). Predicting soil carbon loss with warming. Nature, 554(7693), E4.

---

## Referee Comment (RC5) · M. Mayes (Referee) · 16 Mar 2018

This is a very interesting study in which the newly-derived Q10 values from the Crowther et al. 2016 study are incorporated into a post-hoc simulation of 20 CMIP5 Earth system models. Surprisingly, better-constrained data does not reduce uncertainty in predicted soil organic carbon (SOC) values. Rather, predicted SOC stocks are considerably less in comparison, while retaining very large uncertainties. The study seems very well-done, relies on important new findings, uses an important suite of models, and provides important insights into both predicted SOC and model uncertainty. This paper is a logical outcome of the Crowther et al. 2016 study and will certainly be of interest to the Biogeosciences community.

[Figure]

The authors provide a clear description of an assumption regarding metastable SOC stocks (Eqns 2-4), and how stock differences between two soils, or control and warming, can be used in the model. Authors also completely describe the implications of the (necessary) simplifying assumption of a one-pool model. These are both rather sticky subjects and the authors do an adequate job of addressing these concerns.

Specific comments P1 L22: unclear meaning P2 L20: probably could remove statement about a wide range of typical Q10s. P4 L3: replace with "an inverse with all positive entries" P4 L9: "constructed" P6 L8: are these shown? P8 L10: some edits needed P10 L22: unclear what you mean by "models should increase their variability"

---

## Referee Comment (RC6) · Anonymous Referee #5 · 23 Mar 2018

This study was unusual in using a Q10 function derived from changes in soil C pools to model C fluxes. As such, it was a novel contrast with- and useful comparison to earlier studies based on respiration metrics. The modeling rationale is solid and the simplification of the various model formulations to aggregated responses was reasonable. Overall, I'm impressed with the logic and thoroughness of this study.

This work is important for a several reasons. First, it uses a novel derivation of Q10 to address the important topic of soil C dynamics. Second, the reduction in structural complexity of several different models demonstrated how such aggregations could be done comparably and generated a range of predictions based on existing ESMs. Finally, it suggested that more attention to underlying uncertainties in factors controlling C dynamics might improve outcomes – perhaps as a logical alternative to broad data

integration projects.

Regardless of the novelty of this study, the variation associated with the final output is so large that mean estimates of soil C are not different from earlier work, considering the 95% CIs. This simple fact shifts the main focus from differences between these estimates to their similarities, and as the authors noted, reasons why the variation is so large.

The authors raise several points about their underlying assumptions, some also raised by reviewers, noting that uncertainties in soil C stock data, moisture variations, the assumption of steady-state C pool dynamics, uniform temperature sensitivity of various C pools, nutrient limitations, etc., likely all contribute to variations in prediction. Moreover, aggregations across time, space, and structural resolution of both the models and C pools sacrifice fine scale dynamics that are often non-linear and cannot be averaged across coarser scales, e.g., moisture responses of dry-land systems. So, it's not surprising that the variation in output was large.

I recommend publishing this article but given the large uncertainty in final predictions, I also recommend a more thorough discussion of the limitations of such broad scale approaches. I'd like to hear more from the authors about how different sources of variation could be elucidated and addressed to improve model performance.

---

## Author Comment (AC1) · 26 Mar 2018

Thank you for your contributions in this and previous reviews. We hope that the below will address the concerns raised in this review.

Regarding the Q10 lower boundary of 0.1: While we agree that this is an unusual choice we had two reasons for choosing this boundry. First given that we do not disentangle moisture effects, it was conceivable that an increase in soil temperature could result in a decrease in respiration visa-vi a drier soils imposing stronger moisture limitations. Secondly from a numerical prospective choosing a boundary slightly outside the expected numerical range can demonstrate a robust convergence. We will add these justifications to the methods section and hope they satisfy your concerns.

[Figure]

Regarding specifying soil vs air temperature: You are entirely correct and we apologize for letting this slip past us from your previous reviews. We will add 'soil' to each mention of temperature in the manuscript.

Regarding the coarse woody debris pool: We will remove cCwd from the carbon pools and updated the manuscript. There were no significant changes to the results.

Regarding HadGEM2 temperature function: Thank you! We will update the table.

Regarding the allocation matrix: We agree. We will update the discussion and include comments on how a shift in allocation could affect the analysis.

———————————————————

---

## Author Comment (AC2) · 26 Mar 2018

We thank the reviewer for their excitement about this study and the detail with which they have treated this review. We hope that the below address the concerns raised here.

Regarding the quasi-steady state assumption: Since the reviewer acknowledged the necessity of this assumption in the field data we'll confine our reply to the Earth system model analysis here. We would direct the reviewer to a figure in the SI that regrettably was poorly referenced in the main manuscript. Figure S3 shows the ratio between the annual change in soil carbon stock divided by either the maximum input or output to the soil at the beginning and end of the 21st century. While there are some grid cells in

[Figure]

most models with a relatively high imbalance, the vast majority of models have a ratio below 10%. That is the change in soil carbon stock is at least an order of magnitude less than the input/output fluxes, allowing us to assume that the inputs approximate the outputs and apply equation 3 as stated. We will expand on the numerical justification illustrated in Figure S3 in the methods and would draw the reviewer's attention to the second paragraph in the result section that expands on this point.

The reviewer also expressed concerned about not using a direct Q10 estimate from the heterotrophic respiration. This Q10 estimation from simulated heterotrophic respiration was an approach previously taken in Todd-Brown etal 2014. While we considered this approach for the current study and indeed worked up the analysis and included it in the SI (see Figure S4), we ultimately decided to try to follow the field analysis as closely as possible. The results from this alternate calculation of Q10 did not significantly deviate from the current analysis. We will expand on the implications of Rh vs soil inputs in the methods section.

Regarding the field input assumption: As was correctly pointed out in the manuscript we do address what the implications of changes in inputs would be for this analysis. However given the limitations of the data we are working with this is not possible with the current dataset. We feel we have been upfront about this limitation of the approach and hope that the reviewer can agree that this is a valuable analysis none the less. We plan to address this directly in future studies.

Regarding alternative calculations of the field Q10: The reviewer is entirely correct that we could have chosen to calculate a site by site Q10 instead of the fitted linear regression. We chose the regression approach because of the clear cutoff in Q10 fit at the 2 standard deviation point described in the methods and familiarity of the field with this statistical approach. We've added a comment to the methods section to this effect.

Line by line:

P1L25: We will 'stimulate' for simulate.

P1L27: We respectfully disagree and have kept this in for clarity.

P3L20: Thank you for catching this. We have written the equation to reflect the version in the SI where we use the form '$(Q\_10*K*A)*C(t)$'. This does not change the results of the analysis.

P3L20: We will substitute 'scalar' for scaler.

P4L2: We will substitute 'vice-versa' for vis-versa.

P6L16: We will substitute 'data-driven' for data-drive

P6L18: We will substitute cell for cells.

P8L10-11: We will replace this topic sentence and rework this paragraph as several reviewers found this confusing. A draft topic sentence now reads: "Typically extrapolating decadal trends from annual results is problematic and understandably controversial. However, given the mathematical structure of traditional soil models and the numerical results seen in the Earth system models, it is appropriate in this specific case of inferring long term temperature sensitivities."

P8L16: The citation for Figure S4 is correct. We have revised and split this paragraph to better explain figure 4 which shows both the implications of using heterotrophic respiration instead of soil inputs to calculate Q10, and how those Q10 distributions shift over time.

P8L25: We fixed this typo.

---

## Author Comment (AC3) · 26 Mar 2018

We would like to thank the reviewer for this and their previous reviews. We hope that the below address the reviewers concerns raised here.

We will go through and update Table 1 and the model descriptions to clarify the models which were modified-CENTURY (which are generally cascade models) and models which are direct implementations. We've removed "CENTURY" from the Temperature column of Table 1 and replaced it with a generic function with a citation to the CENTURY paper in the caption. We apologize for the confusion.

The van Gestel manuscript is a fascinating data set that we look forward to addressing directly in future studies. We've added a comment to this effect in the discussion and

to the section where we discuss the statistical power of the dataset.

---

## Author Comment (AC4) · 26 Mar 2018

Thank you for your comments. I believe that your concerns are addressed in the reply to RC4 in AC3.

---

## Author Comment (AC5) · 26 Mar 2018

Thank you for your interest and review of this study. We hope the the below response to your specific comments adequately addresses the concerns you've raised.

P1L22 Thank you for pointing this out. We will revise the sentence.

P2L20 We've chosen to keep this in, mostly to introduce the concept of a 'typical' Q10 value which we come back to later in the manuscript.

P4L3 Thank you for the rewording. We will make the change.

P4L9 Tense will be corrected. Thank you.

P6L8 Yes, we would direct the reviewer to the results section P7L5. However, several

reviewers have had difficulty with this section and we will rewrite P7L5-12 in an attempt to clarify these results.

P8L10 Thank you for pointing out this confusing section. Several reviewers had difficulty with this section and we will rewrite this paragraph.

P10L22 Thank you for pointing this out. We have expanded on this section to attempt to clarify.

---

## Author Comment (AC7) · 2 Apr 2018

Thank you very much for these citations pertaining to the comments addressed above.

---

## Author Response (AR2)

Dear Dr Weintraub,

Thank you again for your attention to this manuscript. As instructed the typo has been fixed and new version uploaded. We look forward to working through any additional copy-edits in the near future.

Sincerely, Kathe Todd-Brown

Dear Dr Weintraub,

27 April 2018

Thank you for your handling of this manuscript. We hope that these revisions address the reviewers concerns with the manuscript.

Some of the changes to the manuscript include. The exclusion of coarse woody debris from the decomposing carbon pool changed some of the exact global totals in a very minor way which did not affect the overall results of the study. Clarification on the boundaries and selection criteria for the Q10 parameter in the methods section. Clarification on the effect of soil inputs vs outputs (no effect in all but one model) as the driving flux to calculate Q10 in the Earth system model study. And finally, a brief expansion of the discussion around alternative mechanisms that need to be considered when interoperating the results of this study.

Below you will find a point by point response to the reviews as well as a mark-up draft of the changes made to the manuscript.

We thank yourself and the reviewers for their thoughtful comments which greatly improved this manuscript and hope that this satisfies the concerns raised.

Sincerely, Kathe Todd-Brown

**Reply to reviewers**

**REVIEWER 1: C.D. Jones (Referee)** chris.d.jones@metoffice.gov.uk Received and published: 2 March 2018 This is an interesting and well written paper addressing a subject area of clear importance and relevance to the readership of Biogeosciences.

The authors clearly articulate the issue of uncertainty in future land-carbon storage, how this responds to climate change and the vital role played of soil organic carbon and its sensitivity to temperature. The results lead to a change in a central estimate, across multi-model results, of soil carbon changes by 2100, although interestingly the data constraint does not reduce the overall uncertainty. This highlights the claim made in the title of "high uncertainty" and leave this area as an outstanding issue for modellers and process community alike to address.

I recommend publication after addressing a few minor concerns as below.

Chris Jones

The paper continues the work and analysis of these authors from the Crowther et al. 2016 Nature paper. The dataset assembled of soil warming plots and changes in soil carbon is clearly very valuable for understanding this key issue of Earth System Modelling. I much prefer the approach here, by comparing with process-based models and trying to constrain their simulations, than I do the direct extrapolation to global scale as per Crowther et al. So overall I like the approach deployed here and have mainly minor comments. There are clearly assumptions and choices made which affect the results - I don't believe these should prevent publication - the authors appear to have been careful to consider these and discuss their implications.

2 specific examples are:

- the issue of timescales - the approach to diagnose Q10 relies on pseudo-equilibrium of the system. It could therefore be potentially dangerous and misleading to compare the outcomes from a Q10 diagnostic of 2-5 years warming with ESM studies of 100 years. I was pleased to see this tested explicitly and the SI shows detailed reasons to believe that this is not a large confounding issue.

- the issue of using a single-pool simple model to recreate and re-scale the responses. Again, the authors appear to have taken potential concerns seriously and present good arguments why their approach is OK. I suspect that using a two-pool model would give different answers (inevitable for modelling!) - but it's not clear that this simplification is inappropriate. Given that prominent ESMs in the past, and still some at CMIP5, still deploy a single soil carbon pool it is reasonable for a simple-model approach to do so (by definition simple models bring clarity at the expense of detail - I feel it is well argued here that the choice is appropriate).

**Thank you for your contributions to this manuscript in this and previous reviews, as well as your supportive comments regarding this revision.**

I have one issue with some of your methodological choices

- you sample q10 values down to 0.1, and in your ESM scaling define "typical" as going down to q10=0.5. It seems VERY odd (unphysical even) to take q10 below 1. Given this is not a linear relationship, you are not looking at taking gradients down towards 0 which represent being flat. Rather q10=1 is where you hit "flat" - i.e. no sensitivity at all to temperature. Once you go below q10=1 then you get an inverse relationship which says that respiration increase as you get COLDER. I'm not aware of any way this could be possible. So I would strongly suggest you take q10=1 as your lower boundary.

While we agree that a Q10 value as low is 0.1 is unusual we had two reasons for choosing this boundary. First given that we do not disentangle moisture effects, it was conceivable that an increase in soil temperature could result in a decrease in respiration visa-vi drier soils imposing stronger moisture limitations. Secondly from a numerical prospective choosing a boundary slightly outside the expected numerical range can demonstrate a robust convergence. We have added these justifications to the methods section and hope they satisfy your concerns.

- please be very careful discussing response to "temperature" - it is always important to specify clearly if this means air- or soil-temperature. It has long been known that the apparent q10 is much lower when calculated as a response to air temp (Raich & Potter (1995, GBC) suggest a Q10 value of 2.0 for soil temperature is roughly equivalent to 1.7 for air temperature). Please check especially the Bond-Lamberty paper/comparison - they calculate a low Q10, as you say - but isn't this for air-temp? The comparison is then perhaps misleading

**The reviewer is entirely correct and we apologize for letting this slip past us from previous reviews. We have add 'soil' to each mention of temperature in the manuscript.**

- be careful using cCwd - is this a distinct carbon pool? or is it a sub-component of cLitter? I believe that the total vegetation system is captured by cVeg+cLitter+cSoil (e.g. see Jones et al., 2013, J.Clim). Then, within this, cLitter is split into tier-2 variables of cCwd, cLitterAbove, cLitterBelow - these are intended to allow reporting in greater detail, but are not new pools. So I think you should remove any cCwd data from your study. I appreciate this is not well explained in the CMIP5 data request - we tried to clarify this for CMIP6 - see figure 5 of Jones et al (C4MIP documentation paper, GMD, 2016)

**We removed cCwd from the carbon pools and updated the manuscript. There were minor changes to the totals in the CCSM, CESM and NorESM1 models but no significant changes to the results.**

- Although HadGEM2-ES does base it's soil carbon scheme on RothC it is not identical - in particular we chose to keep a uniform q10=2 rather than the RothC temperature function (which we found to be not well behaved at very low T). Otherwise your description of HadGEM2 is correct.

**Thank you! We have updated the table.**

- the discussion correctly discusses the possible role of soil moisture as a rate modifier. Also to consider are vegetation cover, soil quality, soil structure and changes in input quality. Some models, like RothC, change their decomposition according to overlying vegetation. They also change their allocation (your "b" matrix in equation 1) according to vegetation type and lability of litter inputs. I don't think these are major factors, but worth mentioning its not just T and moisture which change the respiration.

**We agree. We have added a sentence at the end of the discussion on how this would affect the analysis.**

**Anonymous Referee #2 Received and published: 12 March 2018**

This is a great study, in which Todd-Brown et al. creatively combined experimental soil warming data with Earth system model outputs to illustrate the observation informed uncertainty in future responses of soil organic carbon to warming. I'd recommend publication after addressing a few issues outlined below.

**We thank the reviewer for their excitement about this study and the detail with which they have treated this review. We hope that the below address the concerns raised here.**

1. While it's hard to move away from steady state assumption (it leads to very convenient mathematical forms), I think it is important to at least illustrate the bias that steady state assumption can have on estimated Q10. It would be hard to do for observed SOC responses to warming without very crude assumptions about base turnover rate, but seems realistic for the ESM output. The term on the left side of equation 1 can be extracted from the ESM output and used instead of 0, and baseline k as well as Q10's can be reverse-engineered. I think the steady state bias could

be quite substantial, because the 10% (or even 1%) difference between C input and output can have a cumulative effect over time. With observation-derived Q10 estimates, I suggest excluding sites that are with high degree of certainty are not in steady state. An example is Delta Junction site in Alaska, which experienced stand replacing fire in 1999. Even though warming experiment lasted for 10 years it takes many decades for the SOC to recover after a stand replacing fire (Fu et al., 2017), so equation 4 would not be appropriate for this site.

Since the reviewer acknowledged the necessity of this assumption in the field data we'll confine our reply to the Earth system model analysis here. We would direct the reviewer to figure S3, which shows the ratio between the annual change in soil carbon stock divided by either the maximum input or output to the soil at the beginning and end of the 21st century. While there are some grid cells in most models with a relatively high imbalance, the vast majority of models have a ratio below 10% with over half of the grid cells being below 0.1%. Thus the change in soil carbon stock is at least an order of magnitude less than the input/output fluxes, allowing us to assume that the inputs approximate the outputs and apply equation 3 as stated. While the reviewer is absolutely correct that small imbalances in the input/output fluxes can have a large cumulative influence over time, we feel that we have demonstrated that most of the change in the grid-by-grid soil carbon stocks is attributed to shifts in the quasi steady state not in the relaxation of the system to a quasi-steady state. We have expanded on the implications of this figure in the results section.

2. This leads me to equation 4. I would strongly suggest not omitting the ratio of control inputs to warmed inputs. As authors rightly pointed out, if ratio is less than 1, Q10's are going to be underestimated, and if warming leads to decrease in NPP, the Q10's would be overestimated. A meta-analysis study by Wu et al. (2011) illustrated that warming increases total NPP on average by 15%, and belowground NPP by 52%, which has very substantial implications for Q10 (please see the attached image illustrating the effect of change in productivity on Q10's for Delta Junction, AK). I think looking up changes in NPP estimates for every site would make the observation-based Q10 estimates more defendable.

**While we agree with the reviewer that a site by site shift in NPP would be more robust, this is beyond the current scope of the study. We would draw the reviewer to the caveats stated in the discussion around the shifts in inputs. We hope to revisit this in future studies.**

3. Lastly it wasn't completely clear to me why authors chose to estimate observation- based Q10's with the method described in section 2.3 instead of directly solving for it using equation 3 or 4. Could you please touch on that in the updated version of the manuscript?

We have expanded on our reasoning in the methods section. Briefly, by selected the parameter value based on the model-data fit we have a robust way to describe the uncertainty associated with the parameter as well as clearly community the effects of that parameter on the model fit. While the reviewer is absolutely correct that we could have instead described the distribution of the site specific field-Q10 values, we assert that the method used in this study is equally valid.

Below, please see a list of minor issues:

P1L25: "stimulate"?

**P1L25: We will 'stimulate' for simulate.**

P1L27: "(the opposing carbon flux)" can be omitted in my view

P1L27: We respectfully disagree and have kept this in for clarity.

P3L20: It seems that the model is not accurately specified: vector C(t) of size n by 1 cannot be multiplied by n by n matrix K. The accurate specification of the negative term would be "A\*(Q10\*K)\*C(t)"

**P3L20: Thank you for catching this. We have written the equation to reflect the version in the SI where we use the form ' $(Q_{10}*K*A)*C(t)$ '. This does not change the results of the analysis.**

P3L27: "scalar"?

**P3L20: We substituted 'scalar' for scaler.**

P4L2: "vice-versa"?

**P4L2: We substituted 'vice-versa' for vis-versa.**

P6L16: "data-driven"?

P6L16: We substituted 'data-driven' for data-drive P6L18: "grid cell"

**P6L18: We substituted cell for cells.**

P8L10-11: Please revise the topic sentence, it's not very clear.

**P8L10-11: We will replace this topic sentence and rework this paragraph as several reviewers found this confusing.**

P8L16: did you mean Figure S3?

**P8L16: The citation for Figure S4 is correct but we have added a reference to S3 as well. P8L25: typo: "ESM-Q10"**

**P8L25: We fixed this typo.**

P.S. here are the references used in the review:

Fu Z, Li D, Hararuk O, Schwalm C, Luo Y, Yan L, Niu S (2017) Recovery time and state change of terrestrial carbon cycle after disturbance. Environmental Research Letters, 12, 104004.

Wu Z, Dijkstra P, Koch GW, Peñuelas J, Hungate BA (2011) Responses of terrestrial ecosystems to temperature and precipitation change: a meta-analysis of experimental manipulation. Global Change Biology, 17, 927-942.

**Anonymous Referee #3 Received and published: 14 March 2018**

This is a well-written and original paper with findings that are relevant to the Biogeosciences community. The code documentation is particularly great.

I have reviewed for an earlier version of this manuscript, and they have successfully addressed my previous comments, adding some discussion of how a systematic change in input rates (which I understand is not in the dataset but is nevertheless likely) would affect carbon stocks and Q10, and justifying the use of a one-pool model. From the other referee comments, these

points may warrant some additional discussion. Therefore, I recommend publication after minor revision.

**We would like to thank the reviewer for this and their previous reviews. We hope that the below address the reviewers concerns raised here.**

Two additional thoughts: Most versions of the Century soil model embedded in a number of the ESMs listed as 'cascade' models (e.g., the CLM family models) are feedback models where some soil C is transferred from the slow and passive to the active pool. C1

These versions of Century may well be simplified to the 'cascade' model type, but that development is not clear from the descriptions of the ESM soil models.

We have updated Table 1 and the model descriptions to clarify the models which were modified-CENTURY (which are generally cascade models) and models which are direct implementations. We also added a new model table to the SI which fully documents the nested model references to the best of our ability. We've removed "CENTURY" from the Temperature column of Table 1 and replaced it with a generic function with a citation to the CENTURY paper in the caption. We apologize for the confusion.

I know it's hot of the press, but because the authors base their analysis off of the Crowther et al. 2016 dataset, they could also mention the potential implications of the expansion on their analysis in the Van Gestel et al. comment.

van Gestel, N., Shi, Z., van Groenigen, K. J., Osenberg, C. W., Andresen, L. C., Dukes, J. S., ... & Reich, P. B. (2018). Predicting soil carbon loss with warming. Nature, 554(7693), E4.

The van Gestel manuscript is a fascinating data set that we look forward to addressing directly in future studies. We've added a comment to this effect in the discussion and to the section where we discuss the statistical power of the dataset.

RZ Abramoff rose.abramoff@gmail.com Received and published: 12 March 2018

This is a well-written and original paper with findings that are relevant to the Biogeosciences community. The code documentation is particularly great.

I have reviewed for an earlier version of this manuscript, and they have successfully ad- dressed my previous comments, adding some discussion of how a systematic change in input rates (which I understand is not in the dataset but is nevertheless likely) would affect carbon stocks and Q10, and justifying the use of a one-pool model. From the other referee comments, these points may warrant some additional discussion. There- fore, I recommend publication after minor revision.

Two additional thoughts: Most versions of the Century soil model embedded in a number of the ESMs listed as 'cascade' models (e.g., the CLM family models) are feedback models where some soil C is transferred from the slow and passive to the active pool. These versions of Century may well be simplified to the 'cascade' model type, but that development is not clear from the descriptions of the ESM soil models.

I know it's hot of the press, but because the authors base their analysis off of the Crowther et al. 2016 dataset, they could also mention the potential implications of the expansion on their analysis in the Van Gestel et al. comment.

van Gestel, N., Shi, Z., van Groenigen, K. J., Osenberg, C. W., Andresen, L. C., Dukes, J. S., ... & Reich, P. B. (2018). Predicting soil carbon loss with warming. Nature, 554(7693), E4.

**Thank you for this and previous reviews of this study. I believe that your concerns are addressed in RC4.**

M. Mayes (Referee) mayesma@ornl.gov Received and published: 16 March 2018

This is a very interesting study in which the newly-derived Q10 values from the Crowther et al. 2016 study are incorporated into a post-hoc simulation of 20 CMIP5 Earth system models. Surprisingly, better-constrained data does not reduce uncertainty in predicted soil organic carbon (SOC) values. Rather, predicted SOC stocks are considerably less in comparison, while retaining very large uncertainties. The study seems very well-done, relies on important new findings, uses an important suite of models, and provides important insights into both predicted SOC and model uncertainty. This paper is a logical outcome of the Crowther et al. 2016 study and will certainly be of interest to the Biogeosciences community.

The authors provide a clear description of an assumption regarding metastable SOC stocks (Eqns 2-4), and how stock differences between two soils, or control and warming, can be used in the model. Authors also completely describe the implications of the (necessary) simplifying assumption of a one-pool model. These are both rather sticky subjects and the authors do an adequate job of addressing these concerns.

**Thank you for your interest and review of this study. We hope the below response to your specific comments adequately addresses the concerns you've raised.**

Specific comments P1 L22: unclear meaning

P1L22 Thank you for pointing this out. We have simplified this sentence to read: "This study demonstrates that data integration should strive to capture the variation of the system, as well as mean trends."

P2 L20: probably could remove statement about a wide range of typical Q10s.

P2L20 We've chosen to keep this in, mostly to introduce the concept of a 'typical' Q10 value which we come back to later in the manuscript. However, we thank the reviewer for their feedback in this matter.

P4 L3: replace with "an inverse with all positive entries"

**P4L3 Thank you for the rewording. We have made the change.**

P4 L9: "constructed"

P4L9 This has been corrected. Thank you.

**P6 L8: are these shown?**

**P6L8 Yes, we would direct the reviewer to the results section P7L5. However, several reviewers have had difficulty with this section and we have rewritten these section to clarify these points.**

P8 L10: some edits needed

**P8L10 Thank you for pointing out this confusing section. Several reviewers had difficulty with this section and we will rewrite this paragraph.**

P10 L22: unclear what you mean by "models should increase their variability"

**P10L22 Thank you for pointing this out. We have expanded on this section to attempt to clarify.**

Anonymous Referee #5 Received and published: 23 March 2018

This study was unusual in using a Q10 function derived from changes in soil C pools to model C fluxes. As such, it was a novel contrast with- and useful comparison to earlier studies based on respiration metrics. The modeling rationale is solid and the simplification of the various model formulations to aggregated responses was reasonable. Overall, I'm impressed with the logic and thoroughness of this study.

This work is important for a several reasons. First, it uses a novel derivation of Q10 to address the important topic of soil C dynamics. Second, the reduction in structural complexity of several different models demonstrated how such aggregations could be done comparably and generated a range of predictions based on existing ESMs. Finally, it suggested that more attention to underlying uncertainties in factors controlling C dynamics might improve outcomes – perhaps as a logical alternative to broad data integration projects.

Regardless of the novelty of this study, the variation associated with the final output is so large that mean estimates of soil C are not different from earlier work, considering the 95% CIs. This simple fact shifts the main focus from differences between these estimates to their similarities, and as the authors noted, reasons why the variation is so large.

The authors raise several points about their underlying assumptions, some also raised by reviewers, noting that uncertainties in soil C stock data, moisture variations, the assumption of steady-state C pool dynamics, uniform temperature sensitivity of various C pools, nutrient limitations, etc., likely all contribute to variations in prediction. More- over, aggregations across time, space, and structural resolution of both the models and C pools sacrifice fine scale dynamics that are often non-linear and cannot be averaged across coarser scales, e.g., moisture responses of dry-land systems. So, it's not surprising that the variation in output was large.

I recommend publishing this article but given the large uncertainty in final predictions, I also recommend a more thorough discussion of the limitations of such broad scale approaches. I'd like to hear more from the authors about how different sources of variation could be elucidated and addressed to improve model performance.

We would like to thank the reviewer for their time and encouraging comments. We agree that other processes and variables would likely provide some predictive power in this and other studies. We have added a very limited review of mechanisms that are currently being considered for model representation in the discussion.

**Field-warmed soil carbon changes imply high 21st century modeled uncertainty**

Katherine Todd-Brown1, Bin Zheng1, Thomas Crowther2

1Pacific Northwest National Laboratory, Richland, WA, 99354, USA

2Institute of Integrative Biology, ETH Zürich, Univeritätstrasse 16, 8006, Zürich, Switzerland

Correspondence to: Katherine Todd-Brown (katherine.todd-brown@pnnl.gov)

**Abstract.** The feedback between planetary warming and soil carbon loss has been the focus of considerable scientific attention in recent decades, due to its potential to accelerate anthropogenic climate change. The soil carbon temperature sensitivity is traditionally estimated from short-term respiration measurements -- either from laboratory incubations that are artificially

- 10 manipulated, or field measurements that cannot distinguish between plant and microbial respiration. To address these limitations of previous approaches, we developed a new method to estimate soil temperature sensitivity (Q10) of soil carbon directly from warming-induced changes in soil carbon stocks measured in 36 field experiments across the world. Variations in warming magnitude and control organic carbon percentage explained much of field-warmed organic carbon percentage (R2=0.96), revealing Q10 across sites of 2.2, [1.6, 2.7] 95% Confidence Interval (CI). When these field-derived Q10 values were
- 15 extrapolated over the 21st century using a post-hoc correction of 20 CMIP5 Earth system model outputs, the multi-model mean soil carbon stock changes shifted from the previous value of 88±153 Pg-carbon (weighted mean ± 1 SD), to 19±155 Pg-carbon with a Q10 driven 95% CI of 248±191 to -95±209 Pg-carbon. On average, incorporating the field-derived Q10 values into Earth system model simulations led to reductions in the projected amount of carbon sequestered in the soil over the 21st century. However, the considerable parameter uncertainty led to extremely high variability in soil carbon stock projections within each
- 20 model; intra-model uncertainty driven by the field-derived Q10 was as great as that between model variation. This study demonstrates that data integration should capture the variation of the system, as well as mean trends.

**1 Introduction**

5

The flux of carbon dioxide between the soil and atmosphere is a major control on atmospheric carbon dioxide concentrations. Warming temperatures, driven by increases in atmospheric carbon dioxide, have the potential to stimulate carbon 25 decomposition, accelerating its release into the atmosphere (Davidson and Janssens, 2006). If this is not counterbalanced by an equivalent increase in primary productivity (the opposing carbon flux) then it has the potential to drive a land carbonclimate feedback that will accelerate anthropogenic climate change. Recent global compilations of data from ecosystem warming experiments lend support to this idea (Carey et al., 2016), suggesting that warming alone could drive a loss of carbon from the upper soil horizons (Crowther et al., 2016). However, these studies addressed the impact of warming in isolation, and

[revised manuscript text omitted]

**9**

[revised manuscript text omitted]